# Suicide Rates, Social Capital, and Depressive Symptoms among Older Adults in Japan: An Ecological Study

**DOI:** 10.3390/ijerph16244942

**Published:** 2019-12-06

**Authors:** Tsuneo Nakamura, Taishi Tsuji, Yuiko Nagamine, Kazushige Ide, Seungwon Jeong, Yasuhiro Miyaguni, Katsunori Kondo

**Affiliations:** 1Department of Advanced Preventive Medicine, Graduate School of Medical and Pharmaceutical Sciences, Chiba University, Chiba 260-0856, Japan; idex_waka@yahoo.co.jp; 2Department of Social Preventive Medical Sciences, Center for Preventive Medical Sciences, Chiba University, Chiba 260-0856, Japan; tsuji.t@chiba-u.jp (T.T.); yuiko.mail@gmail.com (Y.N.); kkondo@chiba-u.jp (K.K.); 3Department of Family Medicine, Tokyo Medical and Dental University, Tokyo 113-8510, Japan; 4Department of Community Welfare, Faculty of Health Sciences, Niimi University, Okayama 718-8585, Japan; k-jeong@niimi-u.ac.jp; 5Department of Gerontological Evaluation, Center for Gerontology and Social Science, National Center for Geriatrics and Gerontology, Aichi 474-8511, Japan; y.miyaguni@ncgg.go.jp; 6Institute for Health Economics and Policy, No.11 Toyo-kaiji Bldg, 1-5-11, Nishi-Shimbashi, Minato Ward, Tokyo 105-0003, Japan

**Keywords:** older people in Japan, depressive symptoms, long-term care needs survey, social support, social participation, suicide rates, suicide countermeasure

## Abstract

Depression is considered the primary risk factor for older people’s suicide. When considering suicide measures, it is necessary to clarify the relationship between depressive symptoms, social capital, and suicide rates. Therefore, we aimed to clarify the relationship between community-level social capital, depressive symptoms, and suicide rates among older people in Japan. We analyzed the data gathered from 63,026 men and 72,268 women aged 65 years and older, totaling 135,294 subjects in 81 municipalities with a population of over 100,000 participants in the 2013 Sixth Long-Term Care Needs Survey and another survey conducted by Japan Gerontological Evaluation Study (JAGES) in 2013 including the same question items as the survey in Japan. Multiple regression analysis revealed that the male suicide standardized mortality ratio (SMR) was positively correlated with depressive symptoms (B = 2.318, *p* = 0.002), and received emotional support (B = −2.622, *p* = 0.014) had a negative correlation with the male suicide SMR. In older males particularly, the received emotional support in the community was independently associated with the suicide rate. Therefore, fostering social support in a community could act as a countermeasure to suicide among older males in Japan.

## 1. Introduction

Suicide is a growing public health concern. According to the World Health Organization (WHO), more than 800,000 people worldwide die from suicide every year with the suicide rate per 100,000 people amounting to 11.4 a year. Although suicide can be prevented, WHO notes the importance of a population-based approach and that comprehensive multi-sector measures contribute to effective national prevention strategies [1].

Depression is said to be the primary risk factor for older people’s suicide [2], and the alleviation of depressive symptoms is considered a method for preventing suicide [3,4]. However, few epidemiological studies have investigated the relationship between depression and suicide, even though suicide is a complex problem influenced not only by individual factors but also by social environment. Recently, social capital has been gaining attention as an important factor involved in the decision to die by suicide [5,6,7]. In the public health field, social capital is defined as “resources that are accessed by individuals as a result of their membership of a network or a group” [8]. Social capital exists both at an individual level and at a community level, each of which can be categorized in terms of cognitive dimensions, such as trust and reciprocity, and structural dimensions, such as social participation and social support [9].

Some previous studies that have examined the relationship between community-level social capital and depressive symptom rates report that there is no relationship [10]. However, others have reported that there is such a relationship [11,12,13,14,15].

Similarly, in previous studies examining the relationship between suicide rates and community-level social capital [16,17,18,19], consensus findings have reported that cognitive social capital, such as trust, is associated with low suicide rates in a community. Conversely, there are a few reports about structural social capital that have indicated that suicide rates are low in areas with high social participation and social support rates. However, it has also been reported that there is no significant correlation between structural social capital and suicide rates [18]. Furthermore, no studies have investigated the relationship between depressive symptoms, structural social capital, and suicide rates.

Additionally, from a public health perspective, we focused on social capital at the community level as a population approach to suicide prevention. Compared with cognitive social capital (such as trust and reciprocity), structured social capital (such as social participation and support) leads to the proposal of concrete intervention.

Therefore, we focused on structural social capital at the community level. We hypothesized that community-level social capital can reduce likelihood of suicide among older people. Additionally, we aimed to clarify the relationship between suicide rates, depressive symptoms, and social capital.

In addition, depression is more common in females than males, and the suicide rate is higher for males than females. This finding suggests gender differences in the path to suicide. Therefore, we analyzed suicide rate by gender.

## 2. Method

### 2.1. Study Design

This research was designed as an ecological study using cross-sectional data gathered from 81 municipalities in Japan.

We used data combined with the sixth Long-Term Care Needs Survey [20] that was conducted by the local government in 2013 to formulate a long-term care insurance plan, and another survey conducted by Japan Gerontological Evaluation Study (JAGES) in 2013 including the same question items as the Needs survey. The JAGES is a cohort study aimed at creating a scientific foundation for preventive policies to create a society with healthy longevity.

Overall, 343,483 people from 165 municipalities contributed data to this study. If the population size of the target population is small, the number of suicide deaths is small, reducing the accuracy of the suicide standardized death ratio. If the accuracy of the suicide standardized mortality ratio is low, the suicide standardized mortality ratio is likely to become extremely large or small, which is difficult to interpret. To ensure a certain level of accuracy, a certain scale of the population is required. Therefore, we calculated a 95% confidence interval for the standardized suicide ratio, and examined the appropriate population size [21]. Based on the results, we excluded municipalities less than 100,000 because of large chance errors.

In the questionnaire, we asked the question, “Do you need some type of care and support in your daily life?” Respondents who indicated that they did not need care or support were considered to live an independent life. Based on this result, we selected the people living independently. 

Excluding those who were dependent for physical and/or cognitive functioning, we analyzed the data gathered from 63,026 men and 72,268 women aged 65 years or older, totaling 135,294 subjects from 81 municipalities. We analyzed data on Needs surveys of 33 local governments and added data from 48 local governments that participated in the JAGES survey. The response rate to JAGES was 71.1%, and although the detailed response rate to the Needs survey conducted by municipalities remains unknown, it was surmised to be about the same degree.

### 2.2. Dependent Variable

Each year, the Japanese government reports the number of suicides and the population of each municipality. We obtained the population and number of suicides from the Ministry of Health, Labor and Welfare [22]. We calculated the suicide rate per 100,000 people by dividing the cumulative number of suicides for three years (i.e., 2012–2014) by the cumulative population for the same period [23]. The objective variable was the age-adjusted standardized mortality ratio (SMR) for the suicide of males and females aged older than 60 years over a span of 3 years (2012−2014) [24]. First, we calculated the actual number of suicides for older people in the relevant municipality over 3 years. Next, we calculated the expected number of suicides by multiplying the suicide rate calculated from the number of suicides and the population of older people in Japan of the same age by the older people population of the relevant municipality. Third, we divided the actual number of suicides by the expected number of suicides to determine the age-adjusted standardized death rate. The age-adjusted suicide rate was calculated based on the standard population in 2010 [25].

### 2.3. Explanatory Variables

We analyzed the questionnaire data and calculated the rate of depressive symptoms as per the basic-checklist criteria [26,27]. The basic checklist was prepared by the Ministry of Health, Labor, and Welfare to evaluate depressive symptoms. This involves a subjective evaluation of the necessity of long-term care for older people. The proportion who were considered to be suffering from depressive symptoms were those who answered “yes” to two or more of the five questions on depressive symptoms [26]. The following five questions were asked: (1) “Do you feel you have not experienced any self-fulfillment in everyday life during the last two weeks?” (2) “Has something you used to enjoy very much become boring in the last two weeks?” (3) “Has something you used to do easily become bothersome during the last two weeks?” (4) “Do you think that you have not been useful in the last two weeks?” (5) “Have you felt tired for no reason during the last two weeks?”

Satake’s paper [27] reported that the evaluation of the basic checklist is significantly correlated with the GDS-15 evaluation of the commonly used indicator of depression. Moreover, Cronbach’s alpha of these five questions in the basic checklist was 0.714. Because we confirmed that the reliability (internal consistency) was sufficiently high, we determined that yes responses to more than two out of the five questions related to depression were depressive symptoms.

We evaluated social participation according to the total score (0–400 points) of participants who reported a minimum of once-per-month participation in a volunteer group, sports group, hobby group, or learning activity, by using the criteria of Saito et al. [12]. Moreover, Cronbach’s alpha of these four types of social participation was 0.817. Because we confirmed that the reliability (internal consistency) was sufficiently high, we used social participation scores that integrated the four items in the analysis. Social support was the fourth variable of received and provided emotional support, as well as of received and provided instrumental support. Received emotional support was determined by the percentage of those who answered the question, “Do you have someone who listens to your concerns and complaints?” in the affirmative. Provided emotional support was determined by the percentage of those who answered the question, “Do you listen to someone’s concerns and complaints?” in the affirmative. Received instrumental support was determined by the percentage of those who answered the question, “Do you have someone who looks after you when you are sick and confined to bed for a few days?” in the affirmative. Provided instrumental support was determined by the proportion of those who answered the question, “Do you look after someone when he/she is sick and confined to a bed for a few days?” in the affirmative.

### 2.4. Covariates

We used the population density of habitable land, the percentage of an education attainment of less than nine years, the percentage of single older people, poor self-rated health, poor economic conditions, and unemployment rate as covariates. We obtained our data on the percentage of older people with nine years educational attainment or less, the population density of habitable land, and the unemployment rate from the database of the Statistics Bureau in Japan [28]. We determined the proportion of those who responded that they lived alone from the survey question about family composition. Furthermore, we used a question about participants’ self-rated health in the questionnaire, “Do you think it is healthy to be by yourself?” (1 = very healthy, 2 = somewhat healthy, 3 = not very healthy, or 4 = not healthy). We evaluated poor health as the percentage of those choosing options 3 or 4. Moreover, we used a question about the participants’ economic condition in the questionnaire, “Do you think your economic condition is poor?” (1 = very poor, 2 = somewhat poor, 3 = not very poor, or 4 = not poor). We evaluated the economic condition of a participant as poor by the percentage of those choosing options 1 or 2.

### 2.5. Statistical Analysis

Explanatory variables and covariates excluding the population density of habitable land, educational attainment, and unemployment rate were calculated using age distribution adjustment with direct standardization using the same standard population as the suicide rate [25]. Spearman’s rank correlation coefficient was calculated to create a correlation matrix table. Multiple linear regression analysis was performed using explanatory variables and covariates with the suicide SMR as the dependent variable.

In Model 1, the percentage of educational attainment of less than 9 years, living alone, poor health, and poor economic condition of the respondent were entered along with the population density of habitable land and unemployment rate. In Model 2, we added the depressive symptoms rate to Model 1. In Model 3, we added social participation to Model 2. In Model 4, we added received and provided emotional support as well as received and provided instrumental support to Model 3.

When the variance inflation factor (VIF) was <10, we judged that there was no multicollinearity [29]. We used IBM SPSS Statistics for Windows, Version 24.0. (IBM Corp., Armonk, NY, USA) as our statistical analysis software. We considered p-values of less than 0.05 (two-sided test) to be statistically significant.

This study was conducted with the approval of the Research Ethics Review Committee of the Nihon Fukushi University (approval number: 10-05, approval date: 30 June 2010) (approval number: 13-14, approval date: 1 July 2013) and the Research Ethics Review Committee of Chiba University (approval number: 2493, approval date: 21 October 2016).

## 3. Results

### 3.1. State of Municiparities to Be Analyzed

Table 1 shows the descriptive statistics of the analytic municipalities. We found that there were regional differences in suicide SMR of approximately 3.5 times for males (from 39.29 to 138.96) and approximately 6 times for females (from 26.11 to 154.98).

### 3.2. Spearman’s Rank Correlation Coefficient

Table 2 demonstrates the results of the Spearman’s rank correlation coefficient. It shows that the male suicide SMR positively correlated with the rate of depressive symptoms (ρ = 0.456), but negatively correlated with social participation (ρ = −0.414), received emotional support (ρ = −0.255), and received instrumental support (ρ = −0.327). Meanwhile, the female suicide SMR positively correlated with the rate of depressive symptoms (ρ = 0.258) and negatively correlated with social participation (ρ = −0.361).

### 3.3. Multiple Regression Analysis of Suicide Rate

Table 3 demonstrates the results of the multiple regression analysis of the male suicide SMR. In Model 1, the results reveal that education (*p* = 0.016) and living alone (*p* = 0.008) were significant predictors of the male suicide SMR (R^2^ = 0.278, adjusted R^2^ = 0.219). In Model 2, the results reveal that education (*p* = 0.008), living alone (*p* = 0.034), and depression (*p* = 0.0004) were all significant predictors of the male suicide SMR (R^2^ = 0.396, adjusted R^2^ = 0.337). In Model 3, the results reveal that depressive symptoms (*p* = 0.002) and living alone (*p* = 0.046) were significant predictors of the male suicide SMR (R^2^ = 0.396, adjusted R^2^ = 0.337). In Model 4, the results reveal that depressive symptoms (*p* = 0.002) and received emotional support (*p* = 0.014) were significant predictors of the male suicide SMR (R^2^ = 0.499, adjusted R^2^ = 0.409). In this model, the maximum value of VIF was 4.118, which did not exceed 10, indicating no multicollinearity.

Table 4 demonstrates the results of the multiple regression analysis for the female suicide SMR. In Model 1, the results reveal that living alone (*p* = 0.021), population density (*p* = 0.024), and poor economic condition (*p* = 0.013) were significant predictors of the female suicide SMR (R^2^ = 0.222, adjusted R^2^ = 0.158). In Model 2, the results reveal that the population density (*p* = 0.024), poor economic condition (*p* = 0.028), and depressive symptoms (*p* = 0.037) were significant predictors of the female suicide SMR (R^2^ = 0.268, adjusted R^2^ = 0.197). In Model 3, the results reveal that population density (*p* = 0.024) and poor economic condition (*p* = 0.026) were significant predictors of the female suicide SMR (R^2^ = 0.270, adjusted R^2^ = 0.188). In Model 4, the results reveal that no variables were significant predictors of the female suicide SMR (R^2^ = 0.319, adjusted R^2^ = 0.197). In this model, the maximum value of VIF was 4.118, which did not exceed 10, indicating no multicollinearity.

## 4. Discussion

This is the first report to investigate the relationship between suicide rate, depressive symptoms, and social capital simultaneously. The results for males demonstrate that social capital is independently associated with suicide apart from depressive symptoms. Until now, early detection, treatment, and intervention for depression have been taken as measures against suicide. However, the results of this study suggest that increasing the number of people involved in social support in the community may result in suicide measures, demonstrating a new direction for measures against suicide.

### 4.1. Relationship between Suicide Rates and Depressive Symptoms

Depression is considered a primary risk factor for suicide in older people [2], but most evidence is based on clinical reports. Epidemiologically, few studies have investigated the relationship between depression and suicide in older people in Japan. This epidemiological study investigates the relationship between the rate of depressive symptoms and suicide SMR among older people in the Japanese community. The screening and treatment of depression are used as suicide countermeasures [30], and it has been reported that screening of depression symptoms and intervention by the community are effective suicide countermeasures. Our results suggest that the male suicide SMR is positively correlated with the rate of depressive symptoms at the community level. We considered that the alleviation of depressive symptoms at the community level could be an effective suicide countermeasure.

### 4.2. Relationship between Community-Level Social Capital and Depressive Symptoms

Opinions in literature have been contradictory, that is, social capital suppresses [11,13,14,31,32] and does not suppress [7,10,33] depression. Our results clarify that social support has a modest effect on depression in males.

### 4.3. Relationship between Suicide Rates and Community-Level Social Capital

Previous studies have reported that older people’s suicide attempts and suicide rates are negatively correlated with community-level structural social capital [34,35,36]. In contrast, Kunst et al. [18] reported that there was no relationship between structural social capital and suicide rates using Congdon’s index as an indicator of structural social capital. This index indicates the proportions of single residents, single-parent households, residents of less than one year, and rented households in a community, and assesses the demographics of these variables. The results demonstrate the population dynamics of the area. However, these results differ from the results of studies of social participation and social support related to suicide rates, because the indicators are different. Studies have suggested that suicide rates are negatively correlated with structural social capital [35,36], suggesting that increased community-level social participation and social support could help prevent suicide. Furthermore, the importance of a population-based approach to aid suicide prevention has been demonstrated [37]. Thus, we consider the potential for communities that foster community-level social participation and social support to also prevent suicide.

### 4.4. Relationship between Community-Level Social Capital, Depressive Symptoms, and Suicide Rates

This study is the first to investigate the relationship between social capital, the rate of depressive symptoms, and the suicide rate in Japanese people over 60 years of age at the same time. By examining the relationship between the three, it becomes clear that depressive symptoms and social capital could be related to suicide. According to the findings in this study, social capital such as received emotional support and depressive symptoms are considered to be independently related to the male suicide SMR.

A previous study investigating how social support is independently related to suicide rates examined suicidal ideation and related factors, and reported that social participation promotes social support and social adaptation through social networks, as well as decreased depression and suicidal ideation. Additionally, they considered social participation to be a basic countermeasure against older people’s suicide, rather than screening for patients, early detection, and the treatment of depression [34].

Our study showed a negative correlation between received emotional support and the male suicide SMR in Model 4. Takizawa et al. [38] reported that social support has a stress buffering effect for men, mainly because men depend on their wives as their biggest support, while many husbands did not seem to be a source of support for their wives.

A longitudinal study reported that participation in an organization was related to a decrease in attempted suicide [36]; thus, we suggest suicide countermeasures should include fostering social capital in the community.

As shown in Table 4, the results of multiple regression analysis of the female suicide rate revealed a positive relationship between depression and suicide until Model 2. Social participation in Model 3 and social support in Model 4 eliminated the significance of depression. Model 4 showed a negative association between population density and the female suicide SMR. The results did not reveal a relationship between female suicide rates, depressive symptoms, and social capital. However, the results indicate that female suicide rates may be high in areas with low population density. Previous research has also shown that reduced accessibility to healthcare and social services is related to suicide [39]. Further, people living in a less densely populated area tend to suffer from depression due to poor access to medical resources [40]. Social capital is also poor in these areas, and is considered to have a weak protective effect on mental health [41]. In women, there was no significant association between suicide rate and social capital and depression rates. In men, the received emotional support was relevant. Many older people receive support from their spouses, and the results of this study indicate that men may be more dependent on emotional support from their spouse compared to women. In addition, it has been reported that women have a higher rate of attempted suicide than men [42]. From this, there may be a possibility that the background leading to suicide is different according to gender, with a difference in the strength of ideation to attempt suicide. In the future, it will be necessary to investigate the factors that lead to suicide and the factors that affect the strength of suicidal ideation by gender.

### 4.5. Strengths and Limitations

This study’s strength lies in our simultaneous evaluation of suicide rates, community-level social capital, and rates of depressive symptoms using self-rating data based on a sample of 135,294 older people living in 81 municipalities throughout Japan. This study is the first ecological study to clarify the relationship between suicide rates and depressive symptoms among people aged 60 years and over in Japan and find that received emotional support alleviates the suicide rate apart from the relief of depressive symptoms. Our evaluation clarified the association between depressive symptoms, structural social capital, and suicide rates among older people in Japan.

However, this study had the following limitations. First, the research could not deduce any causal relationship between suicide rates, depressive symptoms, and social participation and social support, owing to the cross-sectional research design. However, based on previous studies [34,43,44] and the World Psychiatric Association guidelines by Baldwin et al. [45], it appears that social support and social participation contribute to the alleviation of both depressive symptoms and the suicide rate.

Second, the number of suicides in Japan is published for each age group according to 10-year age spans. Therefore, data on the suicide rate for the older people are only published for people aged 60 and older, whereas data for those aged 65 and older are not. As a result, although the data for the suicide rate refers to those over the age of 60, the data for other variables refers to those over 65 years of age, thus revealing a discrepancy.

Third, in this study, community-level social capital was calculated only for respondents, so community social capital may be overestimated.

## 5. Conclusions

The results of this study suggest that there are many male suicides in regions with high rates of depressive symptoms, and that a community approach to relieving depressive symptoms would be a valuable suicide prevention measure. Structural social capital such as social participation and social support appear to be related to the alleviation of depressive symptoms and suicide. In the older male population in particular, received emotional support in the community was independently associated with the suicide rate, even after considering depressive symptoms and potential confounders. Therefore, fostering social support in the community could act as a countermeasure against suicide among older males in Japan. While we were unable to determine the relationship between depressive symptoms, social participation, social support, and female suicide in this study, we would like to continue this research in the future.

## Figures and Tables

**Table 1 ijerph-16-04942-t001:** Descriptive statistics of analytic municipalities (*n* = 81).

Variables	Number	Minimum	Maximum	Mean	S.D
Male suicide SMR	81	39.29	138.96	82.95	21.53
Female suicide SMR	81	26.11	154.98	89.38	26.11
Population density (number of people/km^2^)	81	339.2	18,253.7	6279.40	3999.68
Education < 9 years (%)	81	11.2	61.7	29.12	11.73
Living alone (%)	81	6.2	31.3	17.54	5.23
Poor health condition (%)	81	5.5	23.0	16.34	2.76
Poor economic condition (%)	80	26.7	68.9	45.88	10.70
Unemployment rate (%)	81	4.0	10.5	6.06	1.27
Depression (%)	81	16.1	32.4	24.61	3.19
Social participation (point)	81	46.29	113.57	84.69	12.23
Received emotional support (%)	81	75.7	98.4	93.79	2.64
Provided emotional support (%)	81	83.3	94.7	92.05	1.89
Received instrumental support (%)	81	87.4	98.6	94.46	2.01
Provided instrumental support (%)	81	70.7	89.7	80.92	4.11

**Table 2 ijerph-16-04942-t002:** Spearman’s rank correlation coefficient (ρ) among variables * *p* < 0.05, ** *p* < 0.01.

	Variable	1	2	3	4	5	6	7	8	9	10	11	12	13	14
1	Male suicide SMR	1.000													
2	Female suicide SMR	0.255 *	1.000												
3	Population density	−0.012	−0.249 *	1.000											
4	Education < 9 years	0.236 *	0.311 **	−0.658 **	1.000										
5	Living alone	0.363 **	0.072	0.428 **	−0.219 *	1.000									
6	Poor health condition	0.239 *	0.265 *	−0.302 **	0.333 **	0.179	1.000								
7	Poor economic condition	0.308 **	0.072	−0.494 **	0.373 **	0.245 *	0.458 **	1.000							
8	Unemployment rate	0.231 *	0.101	−0.020	0.223 *	0.441 **	0.204	0.392 **	1.000						
9	Depression	0.456 **	0.258 *	0.164	0.063	0.373 **	0.212	0.115	0.341 **	1.000					
10	Social participation	−0.414 **	−0.361 **	0.417 **	−0.682 **	−0.088	−0.274 *	−0.381 **	−0.307 **	−0.370 **	1.000				
11	Received emotional support	−0.255 *	0.021	−0.226 *	0.067	−0.308 **	0.168	−0.044	−0.236 *	−0.201	0.129	1.000			
12	Provided emotional support	−0.247 *	−0.097	−0.082	−0.134	−0.186	0.057	0.021	−0.146	−0.261 *	0.270 *	0.633 **	1.000		
13	Received instrumental support	−0.327 **	−0.118	−0.557 **	0.345**	−0.714 **	0.061	−0.006	−0.231 *	−0.318 **	−0.068	0.471 **	0.246 *	1.000	
14	Provided instrumental support	−0.135	−0.100	−0.516 **	0.180	−0.262 *	0.213	0.523 **	−0.158	−0.169	0.058	0.327 **	0.377 **	0.386 **	1.000

**Table 3 ijerph-16-04942-t003:** Summary of multiple regression statistics for the male suicide SMR.

Model	Variables	Standardized Coefficients β	Non-Standardized Coefficients B	Significant Probability	Standard Error
1	Population density	0.143	0.001	0.385	0.001
	Education < 9 years	0.343	0.617	0.016	0.249
	Living alone	0.399	1.616	0.008	0.590
	Poor health condition	−0.021	−0.165	0.851	0.875
	Poor economic condition	0.099	0.196	0.454	0.260
	Unemployment rate	0.063	1.040	0.613	2.047
2	Population density	0.144	0.001	0.342	0.001
	Education < 9 years	0.346	0.622	0.008	0.230
	Living alone	0.296	1.198	0.034	0.555
	Poor health condition	−0.062	−0.476	0.559	0.811
	Poor economic condition	0.168	0.334	0.173	0.243
	Unemployment rate	−0.053	−0.878	0.655	1.954
	Depression	0.391	2.595	0.0004	0.694
3	Population density	0.151	0.001	0.322	0.001
	Education < 9 years	0.274	0.494	0.091	0.289
	Living alone	0.282	1.142	0.046	0.561
	Poor health condition	−0.039	−0.304	0.721	0.846
	Poor economic condition	0.148	0.293	0.244	0.250
	Unemployment rate	−0.058	−0.970	0.623	1.965
	Depression	0.358	2.377	0.002	0.756
	Social participation	−0.110	−0.191	0.463	0.258
4	Population density	−0.095	−0.001	0.551	0.001
	Education < 9 years	0.288	0.519	0.061	0.272
	Living alone	0.213	0.864	0.199	0.666
	Poor health condition	0.021	0.166	0.838	0.810
	Poor economic condition	0.155	0.308	0.380	0.348
	Unemployment rate	−0.099	−1.649	0.411	1.993
	Depression	0.349	2.318	0.002	0.731
	Social participation	0.022	0.039	0.890	0.278
	Received emotional support	−0.328	−2.622	0.014	1.044
	Provided emotional support	0.139	1.556	0.227	1.275
	Received instrumental support	−0.140	−1.475	0.368	1.627
	Provided instrumental support	−0.158	−0.814	0.339	0.846

**Table 4 ijerph-16-04942-t004:** Summary of multiple regression statistics for the female suicide SMR.

Model	Variables	Standardized Coefficients β	Non-Standardized Coefficients B	Significant Probability	Standard Error
1	Population density	−0.390	−0.003	0.024	0.001
	Education < 9 years	0.269	0.598	0.066	0.320
	Living alone	0.357	1.788	0.021	0.757
	Poor health condition	0.112	1.066	0.346	1.124
	Poor economic condition	−0.345	−0.847	0.013	0.334
	Unemployment rate	−0.081	−1.659	0.530	2.629
2	Population density	−0.389	−0.003	0.022	0.001
	Education < 9 years	0.270	0.601	0.059	0.313
	Living alone	0.292	1.464	0.057	0.755
	Poor health condition	0.086	0.825	0.457	1.104
	Poor economic condition	−0.301	−0.740	0.028	0.330
	Unemployment rate	−0.153	−3.144	0.241	2.661
	Depression	0.245	2.010	0.037	0.945
3	Population density	−0.384	−0.003	0.024	0.001
	Education < 9 years	0.220	0.490	0.217	0.394
	Living alone	0.282	1.416	0.069	0.766
	Poor health condition	0.102	0.974	0.402	1.155
	Poor economic condition	−0.316	−0.775	0.026	0.340
	Unemployment rate	−0.157	−3.224	0.233	2.681
	Depression	0.222	1.821	0.082	1.031
	Social participation	−0.077	−0.165	0.640	0.352
4	Population density	−0.486	−0.003	0.010	0.001
	Education < 9 years	0.227	0.505	0.203	0.393
	Living alone	0.117	0.588	0.543	0.960
	Poor health condition	0.117	1.117	0.342	1.168
	Poor economic condition	−0.242	−0.593	0.242	0.502
	Unemployment rate	−0.213	−4.371	0.133	2.875
	Depression	0.193	1.582	0.138	1.054
	Social participation	−0.076	−0.163	0.686	0.400
	Received emotional support	−0.026	−0.261	0.863	1.506
	Provided emotional support	0.159	2.207	0.234	1.839
	Received instrumental support	−0.282	−3.682	0.121	2.347
	Provided instrumental support	−0.147	−0.937	0.445	1.220

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
