# Peer review of "Suicide Rates, Social Capital, and Depressive Symptoms among Older Adults in Japan: An Ecological Study"

_ijerph, 2019, doi:10.3390/ijerph16244942_

Round 1
Reviewer 1 Report
Thank you for letting me review this study which deals with a very sensitive and crucial topic and uses a nice dataset. I hope my comments will help the authors to improve their manuscript.
Introduction:
No theoretical background. But it clearly fits into a resilience framework since depression is seen as the risk, social capital as a resource and suicide as the outcome. Furthermore, the authors talk about suicide worldwide, but their analysis is based on data from Japan. Thus, the paper needs a cultural framework. Hypotheses missing since this is not an explorative study.p.1, line 39: 800,000 people in general or elderly? If in general, are there any numbers on elderly people?
p.2 lines 49ff: this is a bit confusing about social capital. First, you say that there are 2 levels (individual & community), with each having 2 dimensions (cognitive and structural). I would like to see examples for each possible combination: individual + cognitive, individual + structural, community + cognitive, community + structural. Second, even though you say this, you handle these levels and dimensions in the remainder of the introduction separately. So, you only talk about community social capital, cognitive social capital or structural social capital. But from how you introduce social capital I would expect that you talk about community-structural social capital or community-cognitive social capital, since you say that the two levels have these two dimensions. So, the dimension would not make sense without the levels to me right now. Third, it seems that the remainder of the introduction after this definition of social capital is only on community social capital. What happened to individual social capital? Why are you not interested in this?
In the discussion you sometimes talk about e.g., community-level structural social capital.
Methods
2.2: What you describe under this point are not measures. Your measures are 2.3, 2.4 and 2.5. This is the sample description. So please re-name and re-structure your methods section.
“As the rate of suicide in a less-populated city is highly variable, we excluded municipalities with a population of less than 100,000”. -> How many were those? And also, I don’t understand the reason for excluding those. You say because of a high variability you excluded these municipalities, but a high variability is something you want. If everybody would be the same then there wouldn’t be anything to explain. And since you say that you do an ecological study you would benefit from as much municipalities as possible. So please elaborate more on this exclusion criteria and you should also provide some references for what you state. Maybe something to include in the introduction.
Going this way you need to frame your introduction to suicide in “big cities”.
line 81: "Excluding those who were dependent for physical and/or cognitive functioning". How did you know this and what does this mean?
2.3: This whole paragraph needs more explanation. I don’t understand why you chose this dependent variable. How does dividing the number of suicides within 3 years by the cumulative population within three years result in the suicide rate per 100000 people? There needs to be some transformation to so that you can say “per 100000 people”. Otherwise it is just suicide rate / absolute population summed up for three years. And why didn’t you just use the total population of each municipality?
And why did you use the cumulative population for three years? If a municipality has 3 suicides over a period of 3 years and 300 people live there each year, this would result in 3/900 = .003. what does this say? It would make more sense if you would use the average population over three years (300), since 900 never lived there. This would show that over three years 1% of the real population committed suicide, 900 never lived there. Is there a reference for what you have done?
- line 87: Population and number of suicides for whom? For each municipality?
- give more detail on what age-adjusted means.
- there is no explanation for why you did this separately for males and females. This needs to be explained in the introduction. So you also have a gender perspective which his essential in all your main variables of interest since it is known that males and females differ in their depression and social capital, and they have different modes of doing suicide.
2.4: reliability is missing for the measure of depression.
So how did your depression variable look in the end? Binary? Depression yes vs. no? Even though you cite Saito, it would still be interesting to have at least one sentence on how this social participation variable is derived. Does this require a reliability analysis?2.5: You have many questionable variables that need some explanation of why they are in there. Why population density of habitable land? Why less than nine years of education (Why not using a continuous variable?)? why percentage of single older people?
2.4 & 2.5: you need to show which of these variables relate to individual cognitive/structural social capital and community cognitive/structural social capital. Otherwise your introduction loses a bit of its meaning. Maybe you do a paragraph just on your social capital indicators.
Results:
Line 153-155: This is redundant information, since it is in the table. What can you say that is not in the table or what can you say from a meta-level about the descriptives?
Regression table: important information is missing: R² for each model, delta R² between each step, standardized beta coefficients, standard error. You especially need standardized betas in order to be able to compare your different models.
The VIF is a precondition in order to do a regression and should thus be reported before you show the actual results. You don’t say anything about the VIF for males.
Discussion:
Much of the discussion reads like an introduction. First, say what your findings mean and then discuss them in the light of previous research.
Line 195-196: What are you interpreting here? I guess your correlations? You should mention that this is only for social participation for both genders. But if you refer to your regressions, than this only is true for males.
Correlations cannot be interpreted in a causal way.
4.2: what is this paragraph about? Are you interpreting any of your results? This reads like a literature review.
4.4 why are depression and none of the social capital variables not a significant predictor of female SMR? Since this was the main question of your study, you should now provide some explanation for this finding.
Lines 273-276: Since you have the data you should include municipalities with less than 100000 people in your revision. Here you even bring up reasons why you should have included them from the beginning
Author Response
Response to Reviewer 1 Comments
Point 1 Introduction:
No theoretical background. But it clearly fits into a resilience framework since depression is seen as the risk, social capital as a resource and suicide as the outcome. Furthermore, the authors talk about suicide worldwide, but their analysis is based on data from Japan. Thus, the paper needs a cultural framework. Hypotheses missing since this is not an explorative study.
p.1, line 39: 800,000 people in general or elderly? If in general, are there any numbers on elderly people?
Response 1 (in red)
Thank you for your comment. We hypothesized that community-level social capital would reduce suicide in older people. Thus, I revised the introduction.
Additionally, the number of suicides was announced by the World Health Organization, and the total number of suicides was approximately 800,000. Because the figures for elderly individuals were not publicly available, we did not know the exact number of elderly people suicides.
Line 66: Additionally, from a public health perspective, we focused on social capital at the community level as a population approach to suicide prevention. Compared with cognitive social capital, such as trust and reciprocity, structured social capital, such as social participation and support, leads to the proposal of concrete intervention.
So, we focused on structural social capital at the community level. And we hypothesized that community-level social capital would reduce suicide in older people. Additionally, we aimed to clarify the relationship between suicide rates, depressive symptoms, and social capital.
In addition, depression is more common in females than males, and the suicide rate is higher for males than females. This finding suggests gender differences in the path to suicide. Therefore, we analyzed the suicide rate by gender.
Point 2: p.2 lines 49ff: this is a bit confusing about social capital. First, you say that
there are 2 levels (individual & community), with each having 2 dimensions (cognitive and structural). I would like to see examples for each possible combination: individual + cognitive, individual + structural, community + cognitive, community + structural. Second, even though you say this, you handle these levels and dimensions in the remainder of the introduction separately. So, you only talk about community social capital, cognitive social capital or structural social capital. But from how you introduce social capital I would expect that you talk about community-structural social capital or community-cognitive social capital, since you say that the two levels have these two dimensions. So, the dimension would not make sense without the levels to me right now. Third, it seems that the remainder of the introduction after this definition of social capital is only on community social capital. What happened to individual social capital? Why are you not interested in this?
In the discussion, you sometimes talk about, for example, community-level structural social capital.
Response 2:
Thank you for valuable comments.
We posit that personal social capital may be associated with suicide, but because suicide is extremely low in incidence, verifying the relevance at the individual level is difficult.
Additionally, from a public health perspective, we focused on social capital at the community level as a population approach to suicide prevention.
Compared with cognitive social capital, such as trust and reciprocity, structured social capital, such as social participation and support, leads to the proposal of concrete interventions.
Therefore we conducted our research by focusing on structural social capital at the community level.
I added this description in the introduction.
Line 66: Additionally, from a public health perspective, we focused on social capital at the community level as a population approach to suicide prevention. Compared with cognitive social capital, such as trust and reciprocity, structured social capital, such as social participation and support, leads to the proposal of concrete intervention.
So, we focused on structural social capital at the community level. And we hypothesized that community-level social capital would reduce suicide in older people. Additionally, we aimed to clarify the relationship among suicide rates, depressive symptoms, and social capital.
Point 3:
2.2: What you describe under this point are not measures. Your measures are 2.3, 2.4 and 2.5. This is the sample description. So please re-name and re-structure your methods section.
Response 3:
We moved the contents of section 2.2 to 2.1 and changed the structure of the method.
Point 4:“As the rate of suicide in a less-populated city is highly variable, we excluded municipalities with a population of less than 100,000”. -> How many were those? And also, I don’t understand the reason for excluding those. You say because of a high variability you excluded these municipalities, but a high variability is something you want. If everybody would be the same then there wouldn’t be anything to explain. And since you say that you do an ecological study you would benefit from as much municipalities as possible. So please elaborate more on this exclusion criteria and you should also provide some references for what you state. Maybe something to include in the introduction.
Response 4:
If the population size of the target population is small, the number of suicide deaths is small, reducing the accuracy of the suicide standardized death ratio. If the accuracy of the suicide standardized mortality ratio (SMR) is low, the suicide SMR is likely to become extremely large or small, which is difficult to interpret. To ensure a certain level of accuracy, a certain scale of population is required.
Therefore, we calculated a 95% confidence interval for the standardized suicide ratio and examined the appropriate population size [21].
The population size is 9000 in City A, 30000 in City B, 100,000 in City C, and 400,000 in City D, and we used the total number of male suicides from 2012 to 2014. Additionally, we calculated the width on one side of the 95% confidence interval for the SMR, and we calculated the SMR as 100.
The results are shown in the table 1.
Table 1
|
City
|
A |
B |
C |
D |
|
Population |
8911 |
29989 |
102381 |
404012 |
|
Numbers of suicide |
5 |
13 |
35 |
146 |
|
95%CI Width on one side |
148.1 |
76.0 |
40.7 |
18.0 |
From the results in the table, we excluded municipalities less than 100,000 because of large chance errors.
[21] Schoenberg BS: Calculating confidence intervals for rates and ratio.
Neuroepidemiology 2: 257-265, 1983.
Line 85: If the population size of the target population is small, the number of suicide deaths is small, reducing the accuracy of the suicide standardized death ratio. If the accuracy of the suicide standardized mortality ratio is low, the suicide standardized mortality ratio is likely to become extremely large or small, which is difficult to interpret. To ensure a certain level of accuracy, a certain scale of the population is required. Therefore, we calculated a 95% confidence interval for the standardized suicide ratio and examined the appropriate population size [21]. Based on the results, we excluded municipalities less than 100,000 because of large chance errors.
Point 5:
line 81: "Excluding those who were dependent for physical and/or cognitive functioning". How did you know this and what does this mean?
Response 5:
Thank you for your helpful comments. We added sentences as follows.
Line 93: On the questionnaire, we asked the question, “Do you need some type of care and support in your daily life?” Respondents who indicated that they do not need care or support were considered to live an independent life. Based on this result, we selected the people living independently.
Point 6:
2.3: This whole paragraph needs more explanation. I don’t understand why you chose this dependent variable. How does dividing the number of suicides within 3 years by the cumulative population within three years result in the suicide rate per 100000 people? There needs to be some transformation to so that you can say “per 100000 people”. Otherwise it is just suicide rate / absolute population summed up for three years. And why didn’t you just use the total population of each municipality?
Additionally, why did you use the cumulative population for 3 years? If a municipality has three suicides over a period of 3 years and 300 people live there each year, this would result in 3/900 = .003. What does this say? It would make more sense if you would use the average population over 3 years (300) since 900 never lived there. This would show that over 3 years, 1% of the real population committed suicide, and 900 never lived there. Is there a reference for what you have done?
Response 6:
Deaths that occur relatively rarely, such as suicide, are likely to be misjudged in a single year assessment. Therefore, we referred to the paper [23] and calculated the SMR from the cumulative number of suicides over 3 years.
We added references to the method.
[23] Steelesmith, D. L., Fontanella, C. A., Campo, J. V., Bridge, J. A., Warren, K. L., Root, E. D. Contextual factors associated with county-level suicide rates in the United States, 1999 to 2016.
JAMA Network Open 2019; 2(9): e1910936
Line 106: We calculated the suicide rate per 100,000 people by dividing the cumulative number of suicides for three years (i.e., 2012–2014) by the cumulative population for the same period [23]. The objective variable was the age-adjusted standardized mortality ratio (SMR) for the suicide of males and females aged older than 60 years or older than 3 years (2012−2014) [24]. First, we calculated the actual number of suicides for elderly individuals in the relevant municipality over 3 years. Next, we calculated the expected number of suicides by multiplying the suicide rate calculated from the number of suicides and the population of elderly individuals in Japan of the same age by the elderly population of the relevant municipality. Third, we divided the actual number of suicides by the expected number of suicides to determine the age-adjusted standardized death rate. The age-adjusted suicide rate was calculated based on the standard population in 2010 [25].
Point 7:
- line 87: Population and number of suicides for whom? For each municipality?
Response 7:
Each year, the Japanese government reports the number of suicides and the population of each municipality.
Line 104: Each year, the Japanese government reports the number of suicides and the population of each municipality.
Point 8:
- give more detail on what age-adjusted means.
Response 8:
The objective variable was the age-adjusted standardized mortality ratio (SMR) for the suicide of males and females aged older than 60 years or older than 3 years (2012−2014)[24].First, we calculated the actual number of suicides for elderly individuals in the relevant municipality over 3 years. Next, we calculated the expected number of suicides by multiplying the suicide rate calculated from the number of suicides and the population of the elderly individuals in Japan of the same age by the elderly population of the relevant municipality. Third, we divided the actual number of suicides by the expected number of suicides to determine the age-adjusted standardized death rate.
[24] Kenneth J. Rothman, Saunder Grrnland., Timothy L Lash., Modern Epidemiology,2012 Lippincott Williams &Wilkins
Line 108: The objective variable was the age-adjusted standardized mortality ratio (SMR) for the suicide of males and females aged older than 60 years or older than 3 years (2012−2014)[24]. First, we calculated the actual number of suicides for elderly individuals in the relevant municipality over 3 years. Next, we calculated the expected number of suicides by multiplying the suicide rate calculated from the number of suicides and the population of elderly individuals in Japan of the same age by the elderly population of the relevant municipality. Third, we divided the actual number of suicides by the expected number of suicides to determine the age-adjusted standardized death rate.
Point 9:
there is no explanation for why you did this separately for males and females. This needs to be explained in the introduction. So you also have a gender perspective which his essential in all your main variables of interest since it is known that males and females differ in their depression and social capital, and they have different modes of doing suicide.
Response 9:
Depression is more female than male and suicide rate is higher for male than female. This suggests gender differences in the path to suicide. Therefore, we analyze the suicide rate by dividing it between men and women.
Line 73 : In addition, depression is more common in females than males, and the suicide rate is higher for males than females. This finding be gender differences in the path to suicide. Therefore, we analyzed the suicide rate by gender.
Point 10:
2.4: reliability is missing for the measure of depression.
So how did your depression variable look in the end? Binary? Depression yes vs. no? Even though you cite Saito, it would still be interesting to have at least one sentence on how this social participation variable is derived. Does this require a reliability analysis?
Response 10:
Thank you for your helpful comments. We added sentences as follows.
Line 127: Satake’s paper [27] reported that the evaluation of the basic checklist is significantly correlated with the GDS-15 evaluation of the commonly used indicator of depression. Moreover, Cronbach’s alpha of these five questions in the basic checklist was 0.714. Because we confirmed that the reliability (internal consistency) was sufficiently high, we determined that yes responses to more than two out of the five questions related to depression were depressive symptoms.
Line 132: We evaluated social participation according to the total score (0–400 points) of participants who reported a minimum of once-per-month participation in a volunteer group, sports group, hobby group, or learning activity, by using the criteria of Saito et al. [12]. Moreover, Cronbach’s alpha of these four types of social participation was 0.817. Because we confirmed that the reliability (internal consistency) was sufficiently high, we used social participation scores that integrated the four items in the analysis.
Point 11:
2.5: You have many questionable variables that need some explanation of why they are in there. Why population density of habitable land? Why less than nine years of education (Why not using a continuous variable?)? why percentage of single older people?
Response 11:
The educational background and the percentage of living alone were based on Yen Y-C's paper [34]. For educational background, compulsory education is widely practiced in Japan and compulsory for 9 years.
A fairly high percentage of elderly individuals had completed their education in less than 9 years; thus, we examined this proportion of people.
The percentage of living alone was considered a risk factor for suicide and was used for community assessment.
For population density, we referred to Oka [39] and considered the relationship between social capital and population.
Point 12:
2.4 & 2.5: you need to show which of these variables relate to individual cognitive/structural social capital and community cognitive/structural social capital. Otherwise your introduction loses a bit of its meaning. Maybe you do a paragraph just on your social capital indicators.
Response 12:Here, we demonstrated social capital at the community level by aggregating individual social capital. The social capital variables were social participation, received and provided emotional support, and received and provided instrumental support.
Point 13:
Line 153-155: This is redundant information, since it is in the table. What can you say that is not in the table or what can you say from a meta-level about the descriptive?

Response 13:
Thank you for your advice.
We deleted the part you noted.
We removed these sentences.
Lines 153–155: Table 1 shows the descriptive statistics. It indicates that the male suicide SMR ranged from the minimum value 39.29 to the maximum value 138.96 (average value 82.95), and the female suicide SMR ranged from the minimum value 26.11 to the maximum value 154.98 (average value 89.38).
Point 14:
Regression table: important information is missing: R² for each model, delta R² between each step, standardized beta coefficients, standard error. You especially need standardized betas in order to be able to compare your different models.
Response 14:
We present the data of standardized coefficient β, the standard error, in Tables 3 and 4. In addition, the R2 data for each model is described.
Table 3 and Table 4
Line 193: In model 1, the result revealed that education (p = 0.016) and living alone (p = 0.008) were significant predictors of male suicide SMR (R2 = 0.278, adjusted R2 = 0.219). In model 2, the result revealed that education (p = 0.008), living alone (p = 0.034), and depression (p = 0.0004) were significant predictors of male suicide SMR (R2 = 0.396, adjusted R2 = 0.337). In model 3,the result revealed that depressive symptoms (p = 0.002) and living alone (p = 0.046) were significant predictors of male suicide SMR (R2 = 0.396, adjusted R2 = 0.337). In model 4, the result revealed that depressive symptoms (p = 0.002) and received emotional support ( p = 0.014) were significant predictors of male suicide SMR (R2 = 0.499, adjusted R2 = 0.409). In this model, the maximum value of VIF was 4.118, which did not exceed 10, indicating no multicollinearity.
Line 202: In model 1, the result revealed that living alone (p = 0.021)), population density (p = 0.024), and poor economic condition (p = 0.013) were significant predictors of female suicide SMR (R2 = 0.222, adjusted R2 = 0.158). In model 2, the result revealed that the population density (p = 0.024), poor economic condition (p = 0.028), and depressive symptoms (p = 0.037) were significant predictors of female suicide SMR (R2 = 0.268, adjusted R2 = 0.197). In model 3, the result revealed that population density (p = 0.024) and poor economic condition (p = 0.026)) were significant predictors of female suicide SMR (R2 = 0.270, adjusted R2 = 0.188). In model 4, the result revealed that no variables were significant predictors of female suicide SMR (R2 = 0.319, adjusted R2 = 0.197). In model 3, the result revealed that the population density (p = 0.024) and poor economic condition (p = 0.026)) were significant predictors of female suicide SMR (R2 = 0.270, adjusted R2 = 0.188). In model 4, the result revealed that no variables were significant predictors of female suicide SMR (R2 = 0.319, adjusted R2 = 0.197). In this model, the maximum value of VIF was 4.118, which did not exceed 10, indicating no multicollinearity.
Point 15:
The VIF is a precondition in order to do a regression and should thus be reported before you show the actual results. You don’t say anything about the VIF for males.
Response 15:
Line 197: In this model, the maximum value of VIF was 4.118, which did not exceed 10, indicating no multicollinearity.
Line 214: In this model, the maximum value of VIF was 4.118, which did not exceed 10, indicating no multicollinearity.
Point 16:
Line 195-196: What are you interpreting here? I guess your correlations? You should mention that this is only for social participation for both genders. But if you refer to your regressions, than this only is true for males
Response 16:
This expression only applied to men. From the results of regression analysis, social capital was independently associated with suicide rates in men. We made the correction as follows.
Line 218: The results for males demonstrate that social capital is independently associated with suicide, apart from depressive symptoms.
Point 17:
Correlations cannot be interpreted in a causal way.
Response 17:
We changed the sentence as follows.
Line 233: We considered that the alleviation of depressive symptoms at the community level could be an effective suicide countermeasure.
Point 18:
4.2: what is this paragraph about? Are you interpreting any of your results? This reads like a literature review.
Response 18:
Thank you for your valuable comment. We changed the paragraph as follows.
Line 236: Opinions in literature have been contradictory, that is, social capital suppresses [11,13,14,31,33] and does not suppress [7,10,32] depression. Our results clarify that social support has a modest effect on depression in males.
Point 19:
4.4 why are depression and none of the social capital variables not a significant predictor of female SMR? Since this was the main question of your study, you should now provide some explanation for this finding.
Response 19:
In women, no significant association was observed between suicide rate and social capital and depression rates. In men, received emotional support was relevant. Many elderly individuals receive support from their spouses, and the results of this study indicate that men are supported by their spouses, but men do not support women. In addition, the literature reported that women have more suicide attempts and lower suicide rates than men [42]. Thus, the background that leads to suicide and the strength of ideation to attempt suicide may differ between females and males. Thus, further research should investigate the factors that lead to suicide and that affect the strength of suicidal ideation by gender.
[42] Freeman A, Mergl, R., Kohls, E., Szekely, A. Gusmao, R., Arensman, E., Koburger, N., Hegerl, U., Rummel-Kluge, C. A cross-national study on gender differences in suicide intent. BMC Psychiatry (2017) 17:234
Line 282: In women, there was no significant association between suicide rate and social capital and depression rates. In men, the received emotional support was relevant. Many elderly people receive support from their spouses, and the results of this study indicate that men are supported by their spouses, but men do not support women. In addition, it has been reported that women have more suicide attempts and lower suicide rates than men [42]. From these, there may be a possibility that the background leading to suicide is different between female and male, and the difference in the strength of ideation to attempt suicide. In the future, it will be necessary to investigate the factors that lead to suicide and the factors that affect the strength of suicidal ideation by gender.
Point 20:
Lines 273-276: Since you have the data you should include municipalities with less than 100000 people in your revision. Here you even bring up reasons why you should have included them from the beginning
Response 20:
Thank you for this valuable comment. As aforementioned, suicide rates vary greatly depending on the population of the city. If the fluctuation range is large, a correct evaluation of the suicide rate is difficult. We calculated a 95% confidence interval for the standardized death rate for suicide, and we excluded municipalities with a population of less than 100,000 from the analysis because they have a large range of width on one side of the 95% confidence interval of SMR. Therefore, we targeted municipalities of more than 100,000 people this time.
Reviewer 2 Report
Thank you for the opportunity to review the manuscript “Suicide Rates, Social Capital, and Depressive Symptoms among Older Adults in Japan: An Ecological Study”.
The aim of the paper is to analyse the relationships between suicide rates, community-level social capital and depressive symptoms among older people in Japan.
The manuscript has several positive aspects and it is relevant. However, I believe that some aspects could be improved before proceeding to the publication of the article.
Background
Some of the references related to the relationship between depression and suicide in older people (lines 43-44) are more than 15 years ago. Are there no more recent researches?
Measures
The authors explain that data have been collected from 165 municipalities. However, municipalities with a population of less than 100,000 have been excluded. Perhaps maintaining these municipalities might have allowed them to analyze the impact of the population size. Moreover, this decision affects to the sample size that is not large in relation to the regression analysis performed, in which many explanatory variables are used. This may affect the statistical power of the analysis and would question whether the minimum sample criteria for regression models are met (for example, rule of thumb of 10 cases for each variable).
Explanatory variables
In general, the authors do not present the psychometric properties of the measures used in the study. In fact, several variables have been measured with a single item. Have the authors examined the psychometric properties of the instruments used?
Results
When the authors examine the correlations between the main measures of the study, the magnitude of the effects has been taken into account in addition to the statistical significance. From what size of the effect have the authors considered for their interpretation? (lines 159-162).
With respect to the regression models, when the authors write them, it seems that they were writing correlations. They should change the wording and write these results in terms of explanatory value (line 165 and following; line 178 and following).
Another surprising thing when interpreting the regression models is that they do not take into account the variance explained by each model (adjusted R squared). In fact, they should calculate the increase in the explained variance of each model with respect to the previous one, which would give more information regarding the relevance of the results.
Author Response
Response to Reviewer 2 Comments
Point 1:
Background
Some of the references related to the relationship between depression and suicide in older people (lines 43-44) are more than 15 years ago. Are there no more recent researches?
Response 1 (in red ):
We have added a new example from the literature.
[4] Hewton K., CasanasI. Comabella C., Haw, C., Saunders, K.
Risk factors for suicide in individuals with depression: a systematic review
J Affect Disord 2013 May;147(1-3):17-28.
Line 44: Depression is the primary risk factor for older people’s suicide [2], and the alleviation of depressive symptoms is a method to prevent suicide [3][4].
Point 2
The authors explain that data have been collected from 165 municipalities. However, municipalities with a population of less than 100,000 have been excluded. Perhaps maintaining these municipalities might have allowed them to analyze the impact of the population size. Moreover, this decision affects to the sample size that is not large in relation to the regression analysis performed, in which many explanatory variables are used. This may affect the statistical power of the analysis and would question whether the minimum sample criteria for regression models are met (for example, rule of thumb of 10 cases for each variable).
Response 2:
 If the population size of the target population is small, the number of suicide deaths is small, reducing the accuracy of the suicide standardized death ratio. If the accuracy of the suicide standardized mortality ratio is low, the suicide standardized mortality ratio is likely to become extremely large or small, which is difficult to interpret. To ensure a certain level of accuracy, a certain scale of the population is required. Therefore, we calculated a 95% confidence interval for the standardized suicide ratio and examined the appropriate population size [21].
The population size is 9000 in City A, 30,000 in City B, 100,000 in City C, and 400,000 in City D, and we used the total number of male suicides from 2012 to 2014. Additionally, we calculated the width on one side of the 95% confidence interval for the standardized mortality ratio, and we calculated SMR as 100.
The results are presented in the following table 1.
Table 1
|
City
|
A |
ï¼¢ |
C |
D |
|
Population |
8911 |
29989 |
102381 |
404012 |
|
Number of suicides |
5 |
13 |
35 |
146 |
|
95% CI Width on one side |
148.1 |
76.0 |
40.7 |
18.0 |
Based on the results in the table, we excluded municipalities less than 100,000 because of large chance errors.
In addition, if the assumptions (error uniformity, independence, and regression shape specification) are correct with regard to the variables, then the least square estimator, unlike the maximum likelihood estimator, is an unbiased estimator and is independent of sample size. If the normality is correct, the p-value and confidence interval are likely to be accurate, so we considered number of variables is not a problem.
[21] Schoenberg BS: Calculating confidence intervals for rates and ratio.
Neuroepidemiology 2: 257-265, 1983.
Line 85: If the population size of the target population is small, the number of suicide deaths is small, reducing the accuracy of the suicide standardized death ratio. If the accuracy of the suicide standardized mortality ratio is low, the suicide standardized mortality ratio is likely to become extremely large or small, which is difficult to interpret. To ensure a certain level of accuracy, a certain scale of the population is required. Therefore, we calculated a 95% confidence interval for the standardized suicide ratio and examined the appropriate population size [21]. Based on the results, we excluded municipalities less than 100,000 because of large chance errors.
Point 3:
In general, the authors do not present the psychometric properties of the measures used in the study. In fact, several variables have been measured with a single item. Have the authors examined the psychometric properties of the instruments used?
Response 3:
Regarding the percentage of depression, we used the percentage of people who answered yes to more than two out of the five questions on the basic checklist. Satake’s paper [27] reported that the evaluation of the basic checklist is significantly correlated with the GDS-15 evaluation of the commonly used indicator of depression. Moreover, Cronbach’s alpha of these five questions in the basic checklist was 0.714. Because we confirmed that the reliability (internal consistency) was sufficiently high, we determined that yes responses to more than two out of the five questions related to depression were depressive symptoms.
We evaluated social participation according to the total score (0–400 points) of participants who reported a minimum of once-per-month participation in a volunteer group, sports group, hobby group, or learning activity, by using the criteria of Saito et al. [12]. Moreover, Cronbach’s alpha of these four types of social participation was 0.817. Because we confirmed that the reliability (internal consistency) was sufficiently high, we used social participation scores that integrated the four items in the analysis.
Line 127: Satake’s paper [27] reported that the evaluation of the basic checklist is significantly correlated with the GDS-15 evaluation of the commonly used indicator of depression. Moreover, Cronbach’s alpha of these five questions in the basic checklist was 0.714. Because we confirmed that the reliability (internal consistency) was sufficiently high, we determined that yes responses to more than two out of the five questions related to depression were depressive symptoms.
We evaluated social participation according to the total score (0–400 points) of participants who reported a minimum of once-per-month participation in a volunteer group, sports group, hobby group, or learning activity, by using the criteria of Saito et al. [12]. Moreover, Cronbach’s alpha of these four types of social participation was 0.817. Because we confirmed that the reliability (internal consistency) was sufficiently high, we used social participation scores that integrated the four items in the analysis.
Point 4:
Results
When the authors examine the correlations between the main measures of the study, the magnitude of the effects has been taken into account in addition to the statistical significance. From what size of the effect have the authors considered for their interpretation? (lines 159-162).
Response 4:
If the correlation coefficient is 0.2 or greater, we considered that there was a moderate correlation, and 0.6 or more was a strong correlation. However, in such cases, we had to consider multicollinearity.
Point 5:
With respect to the regression models, when the authors write them, it seems that they were writing correlations. They should change the wording and write these results in terms of explanatory value (line 165 and following; line 178 and following).
Response 5:
Thank you for your advice.
In response to the indication, we changed the text as follows.
Line 193 : In model 1, the result revealed that education (p = 0.016) and living alone (p = 0.008) were significant predictors of male suicide SMR (R2 = 0.278, adjusted R2 = 0.219). In model 2, the result revealed that education (p = 0.008), living alone (p = 0.034), and depression (p = 0.0004) were significant predictors of male suicide SMR (R2 = 0.396, adjusted R2 = 0.337). In model 3,the result revealed that depressive symptoms (p = 0.002) and living alone (p = 0.046) were significant predictors of male suicide SMR (R2 = 0.396, adjusted R2 = 0.337). In model 4, the result revealed that depressive symptoms (p = 0.002) and received emotional support ( p = 0.014) were significant predictors of male suicide SMR (R2 = 0.499, adjusted R2 = 0.409).
Line 202: In model 1, the result revealed that living alone (p = 0.021)), population density (p = 0.024), and poor economic condition (p = 0.013) were significant predictors of female suicide SMR (R2=0.222, adjusted R2 =0.158). In model 2, the result revealed that the population density (p = 0.024), poor economic condition (p = 0.028), and depressive symptoms (p = 0.037) were significant predictors of female suicide SMR (R2=0.268, adjusted R2 =0.197). In model 3, the result revealed that population density (p = 0.024) and poor economic condition (p = 0.026)) were significant predictors of female suicide SMR (R2=0.270, adjusted R2=0.188). In model 4, the result revealed that no variables were significant predictors of female suicide SMR (R2 = 0.319, adjusted R2 = 0.197).
Point 6:
Another surprising thing when interpreting the regression models is that they do not take into account the variance explained by each model (adjusted R squared). In fact, they should calculate the increase in the explained variance of each model with respect to the previous one, which would give more information regarding the relevance of the results.
Response 6:
We showed the adjusted R-squared value.
Table 3
Line 193 : In model 1, the result revealed that education (p = 0.016) and living alone (p = 0.008) were significant predictors of male suicide SMR (R2 = 0.278, adjusted R2 = 0.219). In model 2, the result revealed that education (p = 0.008), living alone (p = 0.034), and depression (p = 0.0004) were significant predictors of male suicide SMR (R2 = 0.396, adjusted R2 = 0.337). In model 3, the result revealed that depressive symptoms (p = 0.002) and living alone (p = 0.046) were significant predictors of male suicide SMR (R2 = 0.396, adjusted R2 = 0.337). In model 4, the result revealed that depressive symptoms (p = 0.002) and received emotional support ( p = 0.014) were significant predictors of male suicide SMR (R2 = 0.499, adjusted R2 = 0.409).
Table 4
Line 202 : In model 1, the result revealed that living alone (p = 0.021)), population density (p = 0.024), and poor economic condition (p = 0.013) were significant predictors of female suicide SMR (R2 = 0.222, adjusted R2 = 0.158). In model 2, the result revealed that the population density (p = 0.024), poor economic condition (p = 0.028), and depressive symptoms (p = 0.037) were significant predictors of female suicide SMR (R2 = 0.268, adjusted R2 = 0.197). In model 3, the result revealed that the population density (p = 0.024) and poor economic condition (p = 0.026)) were significant predictors of female suicide SMR (R2 = 0.270, adjusted R2 = 0.188). In model 4, the result revealed that no variables were significant predictors of female suicide SMR (R2 = 0.319, adjusted R2 = 0.197).
Reviewer 3 Report
Thank you for this very interesting and well-written report. I only have a few minor comments/suggestions as listed below:
Abstract - line 26 - please spell out JAGES
Methods - line 81 - was this the only exclusion criteria? how did you define dependence?
lines 127-128 - the economic condition question is worded a bit awkwardly, would "Do you think your economic condition is poor?" work better?
line 144 - change "two-sides" to "two-sided"
Discussion - line 195-196 - I would suggest changing to "some aspects of social capital are independently associated with suicide" or something similar, as not all variables were significantly correlated, particularly in females
line 198 - unclear what you mean by "fostering social capital may result in suicide measures"
limitations - is it possible that individuals that did not participate in the survey may have less social capital, resulting in selection bias?
Author Response
Response to Reviewer 3 Comments:
Point 1:
Abstract - line 26 - please spell out JAGES
Response 1:
JAGES is an abbreviation for Japan Gerontological Evaluation Study.
Line 27 : Japan Gerontological Evaluation Study
Point 2:
Methods - line 81 - was this the only exclusion criteria? how did you define dependence?
Response 2:
On the questionnaire, we asked the question, “Do you need some type of care and support in your daily life?“ Respondents who indicated that they do not need care or support were considered to live an independent life. Based on this result, we selected the people living independently.
Line 93: On the questionnaire, we asked the question, “Do you need some type of care and support in your daily life?” Respondents who indicated that they do not need care or support were considered to live an independent life. Based on this result, we selected the people living independently.
Point 3:
lines 127-128 - the economic condition question is worded a bit awkwardly, would "Do you think your economic condition is poor?" work better?
Response 3:
Thank you for your advice.
We changed the sentence as you indicated.
line 158 : "Do you think your economic condition is poor?
Point 4:
line 144 - change "two-sides" to "two-sided"
Response 4:
Thank you for your advice.
We changed the word as you indicated.
Line 175: two-sided
Point 5:
Discussion - line 195-196 - I would suggest changing to "some aspects of social capital are independently associated with suicide" or something similar, as not all variables were significantly correlated, particularly in females
Response 5:
Thank you for your advice.
We changed the expression as follows:
Line 218: The results for males demonstrate that social capital is independently associated with suicide, apart from depressive symptoms.
Point 6:
line 198 - unclear what you mean by "fostering social capital may result in suicide measures"
Response 6:
Our results demonstrate that increasing the number of people involved in social support in the community is a countermeasure against suicide.
Line 220: However, the results of this study suggest that increasing the number of people involved in social support in the community may result in suicide measures, demonstrating a new direction for measures against suicide.
Point 7:
limitations - is it possible that individuals that did not participate in the survey may have less social capital, resulting in selection bias?
Response 7:
We calculated community-level social capital for only the respondents; thus, we may have overestimated community social capital.
Thus, this shortcoming is a limitation of this study, and we add the following sentence in the limitations.
Line 309: Third, we calculated community-level social capital for only the respondents; thus, we may have overestimated community social capital.
Round 2
Reviewer 2 Report
The authors have addressed my comments and suggestions adequately in their review.
